# Reciprocal Regulation of Shh Trafficking and H_2_O_2_ Levels via a Noncanonical BOC-Rac1 Pathway

**DOI:** 10.3390/antiox11040718

**Published:** 2022-04-05

**Authors:** Marion Thauvin, Irène Amblard, Christine Rampon, Aurélien Mourton, Isabelle Queguiner, Chenge Li, Arnaud Gautier, Alain Joliot, Michel Volovitch, Sophie Vriz

**Affiliations:** 1Center for Interdisciplinary Research in Biology (CIRB), Collège de France, CNRS, INSERM, PSL Research University, 75005 Paris, France; marion.thauvin@college-de-france.fr (M.T.); irene.amblard1@lms.mrc.ac.uk (I.A.); christine.rampon@u-paris.fr (C.R.); aurelien.mourton@gmail.com (A.M.); isabelle.queguiner@college-de-france.fr (I.Q.); alain.joliot@curie.fr (A.J.); michel.volovitch@ens.fr (M.V.); 2Faculty of Sciences, Université de Paris-Cité, 75206 Paris, France; 3Sorbonne Université, 75006 Paris, France; chenge.li@whu.edu.cn (C.L.); arnaud.gautier@sorbonne-universite.fr (A.G.); 4Processus d’Activation Sélectif par Transfert d’Energie Uni-électronique ou Radiative (PASTEUR), Department of Chemistry, École Normale Supérieure, Université PSL, CNRS, 75005 Paris, France; 5Laboratoire des Biomolécules, LBM, Sorbonne Université, École Normale Supérieure, Université PSL, CNRS, 75005 Paris, France

**Keywords:** redox, H_2_O_2_, Shh, cytoneme, filopodia, morphogene

## Abstract

Among molecules that bridge environment, cell metabolism, and cell signaling, hydrogen peroxide (H_2_O_2_) recently appeared as an emerging but central player. Its level depends on cell metabolism and environment and was recently shown to play key roles during embryogenesis, contrasting with its long-established role in disease progression. We decided to explore whether the secreted morphogen Sonic hedgehog (Shh), known to be essential in a variety of biological processes ranging from embryonic development to adult tissue homeostasis and cancers, was part of these interactions. Here, we report that H_2_O_2_ levels control key steps of Shh delivery in cell culture: increased levels reduce primary secretion, stimulate endocytosis and accelerate delivery to recipient cells; in addition, physiological in vivo modulation of H_2_O_2_ levels changes Shh distribution and tissue patterning. Moreover, a feedback loop exists in which Shh trafficking controls H_2_O_2_ synthesis via a non-canonical BOC-Rac1 pathway, leading to cytoneme growth. Our findings reveal that Shh directly impacts its own distribution, thus providing a molecular explanation for the robustness of morphogenesis to both environmental insults and individual variability.

## 1. Introduction

Hedgehog proteins (Hh in invertebrates and Shh or paralogs in vertebrates, collectively Hhs) are secreted morphogens playing key roles in biological processes ranging from embryonic development, proliferation, adult tissue homeostasis, and cancers [1,2,3]. Signaling by Hhs implies a complex protein journey that includes several steps incompletely characterized by, and possibly depending on physiological context [4,5].

The dual lipidation of Hedgehog ligands (palmitate at their N-terminus, cholesterol at their C-terminus after self-cleavage of the primary translation product) tends to tether them to producing cell membranes, begging the question of their mode of transport to exert their signaling function at a distance from their source [3]. Several mechanisms (not mutually exclusive) were proposed to explain the transport of this morphogen, possibly cooperating to various degrees depending on the tissue or stage: lipoprotein aggregates, exosomes, shedding of lipidated peptides, and transport on extended filopodia called cytonemes [6]. In addition, it was demonstrated that Hhs, similarly to Wnt ligands, are very often endocytosed after primary secretion, the final delivery via exosomes and/or filopodia corresponding to a secondary secretion [4,7,8,9,10].

After reaching their targets, Hhs signal via a variety of pathways, canonical via their Patched (Ptch) receptors, the Smoothened (Smo) GPCR and the transcription factors of the Gli family, as well as non-canonical pathways depending or not on Smo and leading to a large array of nuclear or cytoplasmic effects (notably apoptosis, metabolism conversion, cytoskeleton rearrangement, migration, and guidance) [3,5,7,11,12,13,14,15].

Recently, we observed that the Shh compartmentation was modified by hydrogen peroxide (H_2_O_2_) [16], suggesting that H_2_O_2_ could regulate the Shh secretion process. In addition, preliminary data suggested that Shh controls H_2_O_2_ levels during cell plasticity and tissue remodeling in adults [16,17]. H_2_O_2_ has long been exclusively considered as a deleterious molecule damaging cellular integrity and function. It is now becoming evident that H_2_O_2_ also contributes to bona fide physiological processes, notably through protein cysteine targeting [18,19,20]. This is particularly relevant for H_2_O_2_ produced extracellularly by superoxide dismutase (SOD3)–rapidly re-imported into the cell via specific aquaporins–following superoxide synthesis resulting from the NADPH oxidase (NOX) activation at the plasma membrane [21,22,23].

The physiological role of H_2_O_2_ in living systems has gained increased interest and has been investigated in different models of development and regeneration [24,25,26,27]. Several studies have revealed strong spatio-temporal variation of H_2_O_2_ levels in live animals of various species (fly, nematode, zebrafish, and xenopus) during these processes [16,28,29,30,31,32,33,34,35,36,37,38,39,40,41]. The correlation between mild oxidative bursts and developmental events led to early suggestions of a mechanistic link between them [42,43,44]. As the patterning of a developing embryo relies on the graded activity of morphogens [45,46,47], we decided to precisely determine by which mechanism H_2_O_2_ could regulate Shh trafficking. We set up a quantitative assay to measure the efficiency of each step of Shh’s journey and demonstrate that H_2_O_2_ inhibits Shh secretion but enhances Shh internalization by producing cells and subsequent delivery to target cells. Furthermore, Shh internalization per se enhances endogenous H_2_O_2_ levels via a Rac1/NADPH oxidase pathway that induces filopodia growth, thus regulating Shh trafficking in an H_2_O_2_/Shh feedback loop.

## 2. Materials and Methods

### 2.1. Fish Husbandry and Pharmacological Treatments

Zebrafish were maintained and staged, as previously described [41]. Experiments were performed using the standard AB wild-type strain. The embryos were incubated at 28 °C. Developmental stages were determined and indicated as hours post fertilization (hpf). The animal facility obtained permission from the French Ministry of Agriculture for all the experiments described in this manuscript (agreement No. C 75-05-12). To decrease H_2_O_2_ levels, embryos were incubated in VAS-2870 (NADPH oxidase inhibitor; NOX-i) (100 nM) from Enzo Life Sciences (#BML-El395-0010, Enzo Life Sciences, Inc.; Farmingdale, NY, USA) or an equivalent amount of DMSO as a control for the duration of the time-lapse analysis. To enhance H_2_O_2_ levels, D-Alanine (D-Ala, Sigma-Aldrich, St. Louis, MO, USA #A7377) (10 mM) was injected in zebrafish ventricles one hour prior to H_2_O_2_ levels or filopodia analysis.

### 2.2. Expression Constructs, Permanent Cell Lines, and Fish Transgenic Lines

All recombinant DNA (Appendix A) were prepared by standard cloning methods. Plasmids and sequences are available on request.

Shh ligand was tagged at the position described in [48] (just down Gly198, with the intein cleavage signal aa189–198 being repeated at the C-terminus of the tag sequences) to preserve the ligand function. Tags used were: the fluorescent protein mCherry, the fluorogen-activated peptide YFAST (Plamont, 2015, exRef59), the streptavidin binding peptide (SBP) (Boncompain, 2012, exRef60), the small portion of the split nanoluciferase (HiBiT), and combinations of them.

Stable cell lines (Appendix A) were prepared under Hygromycin selection using the HeLa FlpIn-TREX cell line kindly provided by Stephen Taylor [49] and expressing the tetracycline repressor (TREX, Life Technologies).

Transgenic lines used in this study were: 2.4Shha-ABC:Gal4 (this article); UAS:Igkmb5DAOmCherry (this article); UAS:Igkmb5CATmCherry (this article); UAS:Shh-mCherry, GFP-farn (this article); UAS:HyPer7 (this article); olig2:GFP [50]. The transgenic fish lines were constructed as described [51] using pTol2 derivatives containing the appropriate promoter/enhancer and the SV40 late polyadenylation signal (SVLpA). Shh:Gal4 contains the Gal4 DNA binding domain fused to 2 minimal activator sequences (Gal4BD-FF [52]) inserted between the -2.4Shha promoter and ar-A, ar-B, ar-C Shha enhancers (a kind gift from R. Ertzer [53]). UAS:HyPer7 contains HyPer7 sequence [39] down 5xUAS (derived from [54]). UAS:igkmb5-DAOmCherry and UAS:igkmb5-CATmCherry contain the same enhancer downstream and provided with a signal peptide (Igk) from a kappa light chain and a minimal transmembrane domain (mb5) from CD4, fusions of mCherry (in C-terminal position) with either D-Aminoacid oxidase (DAO [55]) or mouse Catalase deprived from its lysosome-targeting signal (CAT). For bidirectional expression, the 5xUAS regulatory element was flanked by minimal promoter/5′ UTR sequences from pCS2 on one side and CMV on the other side [56].

### 2.3. Embryo Live Imaging and Image Processing

The larvae were anesthetized in tricaine solution and embedded in low-melting agarose (0.8%). Imaging was performed with a CSU-W1 Yokogawa spinning disk coupled to a Zeiss Axio Observer Z1 inverted microscope that was equipped with a sCMOS Hamamatsu camera and a 10× (Zeiss 0.45 Dry WD: 2.1 mm) or a 25× (Zeiss 0.8 Imm DIC WD: 0.57 mm) oil objective. DPSS 100 mW 405 nm and 150 mW 491 nm lasers and a 525/50 bandpass emission filter were used for HyPer7 imaging, and DPSS 100 mW 561 nm laser and a BP 595/50 was used for mCherry imaging. Floor plate cells imaging was performed using a Zeiss LSM 980–AiryScan 2 confocal equipped with an AiryScan detector GaAsP-PMT and 25× (Zeiss 0.8 Imm WD: 0.57 mm) or a 40× (Zeiss 1.3 Oil DIC (UV) WD: 0.22 mm) oil objectives. DPSS 10 mW 488 nm and 10 mW 561 nm lasers and 517 nm and 610 nm AiryScan emission filters, respectively, were used for GFP and mCherry acquisition. AiryScan SR mode was used and AiryScan-processed images were analyzed. To calculate the HyPer ratio, images were treated as previously described [57]. For filopodia analysis, 48 hpf Tg(2.4Shha-ABC:Gal4-FF) larvae expressing Igkmb5-DAO-mCherry were used for fluorescence acquisition as described above, and one slice of each Zstack was extracted. Slices presenting the maximum filopodia number were selected for FiloQuant processing and analysis [58].

### 2.4. Pharmacological Treatments

To decrease H_2_O_2_ levels, cells were treated with extracellular Catalase (CAT_ext_; Sigma-Aldrich, #C1345, 400 U/mL). To increase H_2_O_2_ levels, cells expressing D-amino acid oxidase (DAO) were treated with 10 mM D-alanine (D-Ala; Sigma-Aldrich, #A7377) before the internalization or secretion assays were performed. To inhibit NOX activity, cells were pre-treated for 1 h with 10 μM VAS-2870 (NOX-i; #BML-El395-0010, Enzo Life Sciences, Inc.; Farmingdale, NY, USA) or an equivalent amount of DMSO as a control. To inhibit Rac1, cells were pre-treated for 6 h with 20 μM NSC23766, a Rac1-inhibitor (Rac1-i; No2161, Tocris). To inhibit Dock release from Elmo, cells were pre-treated for 6 h with 100 μM CPYPP (DOCK-i; No4568, Tocris). To inhibit Shh signaling, cells were pretreated for 24 h with 10 μM cyclopamine (Shh-i; #239803, Millipore).

### 2.5. Quantitative Secretion Assay

Cells (13,000 per well) were plated on 96-well plates (Greiner Bio-one) coated with polyornithine (10 μg/mL). After 10 h, the cells were co-transfected with a plasmid expressing Shh-SBP-HiBiT (or a secreted control, SecGFP-SBP-HiBiT) under doxycycline control and a plasmid constitutively expressing the hook (Strepta, core streptavidin-KDEL provided with a signal peptide). Some of these cells received an additional plasmid constitutively expressing a membrane-bound fusion between mCherry and DAO (Lck-Che-DAO). After 24 h, the medium was removed and cells were incubated with fresh medium containing doxycycline for 2 h (to induce HiBiT fusion expression). Medium was changed and secretion was induced with biotin (100 μM final) and, after the purified large fragment of split nanoluciferase (LgBiT) protein addition to the medium, the luciferase activity was measured 1 h later with a 96-well plate luminometer (Tristar, Berthold) as described in the HiBiT assay kit (Promega). The cells were then lysed to measure the intracellular protein expression. Normalization with biotin-untreated wells enabled us to calculate the secretion index and report the secretion efficiency.

### 2.6. Quantitative Internalization Assay

Cells (90,000 per well) stably expressing LgBiT were plated in 24-well culture dishes. After 24 h, the medium was changed and cells were incubated for 30 min with the medium containing HiBiT fusions (taken from cells expressing Shh-HiBiT or secreted mCherry, SecCh-HiBiT for 48 h, and adjusted on protein Luciferase activity), before incubating the cells with trypsin and removing them. After centrifugation, cells were lysed and the luciferase activity of endocytosed protein was measured with a 96-well plate luminometer (Tristar, Berthold) with a HiBiT assay kit (Promega).

### 2.7. Quantitative Intercellular Transfer Assay

Cells (90,000 per well) were plated in 24-well culture dishes. After 10 h, the cells were co-transfected with a plasmid expressing Shh-SBP-HiBiT under the control of doxycycline and a plasmid constitutively expressing Lck-Che-DAO. After 24 h, cells were removed using trypsin and co-cultured on 96-well plates (Greiner Bio-one) coated with polyornithine (10 μg/mL) with cells stably expressing LgBiT anchored to the extracellular side of the cell surface (siL-LgBiT-mb5). After 5 h, cells were incubated with fresh medium containing doxycycline for 2 h (to induce Shh-SBP-HiBiT expression). Medium was then changed and secretion was induced with biotin (100 μM final), and luciferase activity was measured every 1 h over 4 h with a 96-well plate luminometer (Tristar, Berthold) as described in the HiBit assay kit (Promega).

### 2.8. H_2_O_2_ Imaging with the HyPer Probe in HeLa Cells

HyPer (H_2_O_2_ ratiometric probe) fluorescence was excited with DPSS 10 mW 488 nm and 10 mW 405 nm lasers, and the corresponding YFP emission was measured using a 525/50 bandpass emission filter. Spinning-disk images were acquired using a 63× objective (63×/1.4 oil WD: 0.17 mm) on a Spinning-Disk CSU-W1 (Yokogawa) equipped with a sCMOS Hamamatsu 2048 × 2048 camera. To calculate the HyPer ratio, images were treated as previously described [57]. 

### 2.9. Quantification and Statistics

Data were analyzed using GraphPad Prism 6 and expressed as the mean ± standard error of the mean (SEM). Data distribution was first checked for normality. Comparisons between two unpaired groups were performed using the Student’s t-test. For multiple conditions, ordinary one-way ANOVA followed by Tukey’s multiple comparison test or two-way ANOVA followed by Sidak’s or Tukey’s test were performed to evaluate the significant differences. For filopodia analysis, statistical errors (SD) were estimated as √*p*(1−*p*)/*n*, where *p* is the percentage in a class and *n* is the total number investigated (or √1/*n* when *p* = 0 or 1). The degree of significance was represented as follows: * *p*-value ≤ 0.05; ** *p*-value ≤ 0.01; *** *p*-value ≤ 0.001; and **** *p*-value ≤ 0.0001. Sample sizes and number of replicates are given in Appendix A.

## 3. Results

### 3.1. H_2_O_2_ Affects Shh Trafficking in HeLa Cells

To analyze the mechanism by which H_2_O_2_ levels could affect Shh trafficking, we used a cellular system enabling accurate quantification of this process. HeLa cells were chosen to avoid complex feedback loops: these cells do not express endogenous Shh (nor the other two members of the family: Indian Hh, Ihh or Desert Hh, Dhh) or respond to the canonical Shh signaling pathway, but otherwise express pathway components known to be involved in Shh trafficking (lipidation, access to the extracellular space, endocytosis, and delivery to receiving cells, Appendix A). Shh trafficking from producing to receiving cells is a circuitous journey. While some aspects are still a matter of debate, there is no doubt that cis-endocytosis in producing cells can be part of the process [4,7]. Making use of Shh constructs (mouse sequence) tagged according to [48] to preserve the qualitative properties of the protein, we first verified that HeLa cells recapitulated the overall traffic of Shh (Figure 1A–D). To distinguish different steps of the Shh journey (Figure 1A), we exploited a double tag combining a fluorogen-activating peptide (YFAST) and a classical fluorescent protein, the monomeric Cherry (mCherry). Contrary to mCherry, in which chromophore takes time to mature, YFAST fluoresces instantly after the fluorogen addition [59], allowing Shh detection from the beginning of its journey in the endoplasmic reticulum (ER) (step 1 in Figure 1A), which is too early for mCherry detection. However, YFAST is sensitive to pH and cannot fluoresce in endosomes (step 2 in Figure 1A), contrary to mCherry. When finally sent for the receiving cells (step 3 in Figure 1A) Shh should be detectable via both tags. The transfection of HeLa cells with such a construct indeed confirmed the usefulness of this cell line for our purposes (Figure 1B). At a steady state, the majority of cells display diffuse green staining, including the ER, as well as large red spots about the size of endosomes (Figure 1B). HeLa cells with cellular protrusions associated with yellow puncta were found, indicating detection via both tags of Shh en route to receiving cells (Figure 1B’). The end of the journey was also easy to image by co-culturing transfected and untransfected HeLa cells. As shown in Figure 1C,C’, when transfected and untransfected cells were nearby, the ShhmCherry could be detected in the untransfected cells, and the arrangement of cytoplasmic protrusions between the two cells suggested a transfer occurring via filopodia.

To reach an adequate level of precision in the analysis of the H_2_O_2_ effect on Shh trafficking, we needed to synchronize its secretion. We thus made use of the Retention Using Selective Hooks (RUSH) system [60], where Shh, fused to the streptavidin-binding peptide (SBP), is retained in the ER of cells expressing streptavidin fused to the KDEL retrieving signal until the addition of biotin. As shown in Figure 1D, before the biotin addition (t = 0), Shh-SBP-AFP (Shh-SBP is fused to an AutoFluorescent protein, mCherry, here) was hooked in the ER, and the biotin addition allowed us to calibrate the timing of the Shh journey using real-time imaging. It takes approximately 30 to 40 min for the bulk of Shh to reach the Golgi (very similar to many secreted proteins in these conditions, e.g., Wnt3a [61]), secretion is abundant from 1 h, localization in endosomes is conspicuous between 2 and 3 h, and detection in (or at the surface of) non-producing cells lasts up to 5 h and then disappears. HeLa cells thus display typical features of Shh trafficking and can be used to analyze the potential effects of modifying H_2_O_2_ levels if we have the means to rigorously quantify Shh in different compartments at different steps of the process.

In addition, we recently developed [62] quantitative assays to track the different steps of a protein journey by combining the RUSH [60] and HiBiT systems [63] (HiBiT assay, Promega) (Figure 2). The HiBiT system is constituted by a split luciferase: when in the same compartment, the two luciferase moieties, HiBiT (Sbi in plasmid names) and LgBiT (G/Lbi in plasmid names) may spontaneously assemble and restore the luciferase activity that can be measured by the substrate addition (Figure 2B,D,F). These assays are inducible, quantitative, and specifically adapted to protein trafficking. We then set up this quantitative assay to separately analyze each step of Shh’s journey. This strategy allowed us to determine the optimal time frame for analyzing each step of Shh trafficking in cell culture, i.e., secretion (Figure 2C), endocytosis (Figure 2E), and delivery (Figure 2G). These quantitative results are consistent with our direct fluorescence microscopy observations (Figure 1D) and were combined with redox modulation tools to test the hypothesis of the redox regulation of Shh trafficking.

First, we studied the effects of H_2_O_2_ on the primary Shh secretion. To increase H_2_O_2_ levels, we expressed inducible D-aminoacid oxidase (DAO), which produces H_2_O_2_ in the presence of D-Alanine (D-Ala) and is not expressed in HeLa cells [41,55,64]. For cells expressing DAO and treated with 10 mM D-Ala, we observed a specific reduction in Shh secretion, not observed with a secreted GFP (secGFP) construct taken as a control (Figure 3A,B). This result is in close agreement with our previous observations: after oxidative treatment, cells showed reduced Shh secretion, and a pool of Shh was trapped in the Golgi apparatus [16]. Conversely, reduction of H_2_O_2_ levels by the direct addition of Catalase (CAT_ext_) in cell culture [41], enhanced Shh but not SecGFP secretion (Figure 3A,C). Next, we applied the same treatments to study Shh endocytosis on cells incubated with the conditioned media of Shh-expressing cells. Compared to the control conditions (secreted mCherry, SecCh, and conditioned media), enhancing H_2_O_2_ levels with DAO (by the addition of D-Ala) stimulated the Shh endocytosis (Figure 3D,E). Conversely, the reduction in H_2_O_2_ levels with CAT_ext_ treatment reduced Shh endocytosis (Figure 3D,F). Finally, we studied the effects of H_2_O_2_ level modulation on Shh delivery to recipient cells in the co-culture assay. Cells expressing DAO and treated with D-Ala (but not untreated cells) demonstrated increased Shh delivery to recipient cells (Figure 3G,H), while a reduction in H_2_O_2_ levels with CAT_ext_ had the opposite effect (Figure 3G,I). Altogether, these results indicate that physiological variations in H_2_O_2_ levels impact Shh trafficking in HeLa cells. This raises the interesting possibility that the heterogenous H_2_O_2_ distribution could polarize Shh secretion and endocytosis in vivo. We thus decided to look more carefully at the spatial and temporal variation of H_2_O_2_ distribution in the zebrafish embryo during pattern formation in the central nervous system, and whether its experimental manipulation would modify aspects of this patterning.

### 3.2. H_2_O_2_ Levels Are Dynamic in Time and Space in the Embryonic Spinal Cord

Between 25 and 45 h post fertilization (hpf) in zebrafish, Shh is first expressed in the medial floor plate (MFP) consisting of a single row of cells at the central midline, then extends to the flanking lateral floor plate (LFP), playing an important role in the neuroglial switch [65,66]. Using the improved H_2_O_2_ ratiometric probe HyPer7 [39], we first measured H_2_O_2_ levels in Shh-expressing cells in the spinal cord of live zebrafish embryos during this time window (Figure 4). The quantitative analysis of the H_2_O_2_ signal demonstrated a regular and significant decrease (approximately 25%) in H_2_O_2_ levels in the MFP between 25 and 45 hpf (Figure 4A–C). A close-up view of the MFP demonstrated that, in addition to an overall decrease in concentration, the spatial distribution of H_2_O_2_ levels varies over time (Figure 4D–F). A time-lapse analysis of these cells revealed a dynamic intracellular distribution of the HyPer7 signal between 25 and 45 hpf. At the end of this period, H_2_O_2_ levels are homogeneously spread throughout the cell. Between 25 and 35 hpf, however, a distinct gradient is transiently established from higher concentrations apically to lower concentrations basolaterally, and most marked at 31 hpf (Figure 4E,F). Thus, during this neurogenesis period, when Shh induces oligodendrocyte precursor cells (OPCs), H_2_O_2_ levels decrease over time and exhibit a transiently polarized distribution within MFP cells.

### 3.3. H_2_O_2_ Impacts Filopodial Formation and Shh Cellular Targets in the Embryonic Spinal Cord

Shh can be delivered to the receiving cells via filopodia, and it has been proposed that the number and length of filopodia impact Shh distribution [5,9,67,68,69,70,71]. We used a double-transgenic fish 2.4Shha-ABC:Gal4FF; Shh-mChe<UAS>GFP-farn with bidirectional UAS-driven expression to visualize both Shh and plasma membranes in the MFP. Shh-mCherry was indeed detected along and at the tip of filopodia, as shown at 48 hpf (Figure 5A,B’). To test whether physiological modifications of H_2_O_2_ levels could impact filopodia, we expressed DAO at the level of the plasma membrane in MFP cells (2.4Shha:Gal4; UAS:Igk-mb5-DAO-mCherry), and we injected D-Ala into the spinal cord canal at 48 hpf to enhance H_2_O_2_ levels in Shha-expressing cells (Figure 5C). We then visualized the filopodia using the fluorescence of the membrane-bound mCherry (Figure 5D) and quantified the number of filopodia per cell (Figure 5E) as well as the length of the filopodia (Figure 5F). Interestingly, enhancement of H_2_O_2_ levels induced increases in both the number and the length of filopodia (Figure 5E,F) in vivo, suggesting that mild modifications of H_2_O_2_ levels could modulate Shh functioning in the neural tube. To test this hypothesis, we treated zebrafish larvae (2.4Shha:Gal4; UAS:HyPer7) with NOX-i (NADPH oxidase pan inhibitor) to reduce H_2_O_2_ levels (Appendix A) from 30 hpf to 48 hpf. This reduction in H_2_O_2_ levels induces a decrease in the filopodia number in MFP (Appendix A). We then analyzed the distribution of OPCs in the embryonic spinal cord at 72 hpf in (Olig2:EGFP) larvae (Figure 5G–I). A small reduction in H_2_O_2_ levels (approximately 10%) was sufficient to enhance by a factor of 2 the number of OPC (Figure 5I), known to depend on Shh activity [66,72], without affecting shh expression (Appendix A). Thus, small modifications in H_2_O_2_ levels not only modified the number and length of filopodia in vivo but also altered a Shh-dependent switch.

### 3.4. Shh Regulates H_2_O_2_ Levels and Filopodial Growth via a Non-Canonical Route

After demonstrating that H_2_O_2_ impacts Shh trafficking (Figure 3) and signaling outcome (Figure 5), we wondered whether this was integrated into a larger H_2_O_2_-Shh feedback loop, as first observed during adult zebrafish caudal fin regeneration [17,33]. If this were the case, Shh’s interaction with cells should itself impact H_2_O_2_ levels. We first observed that HeLa cells co-expressing HyPer and Shh-mCherry exhibited higher levels of H_2_O_2_ than cells expressing HyPer and mCherry (Figure 6A,B), suggesting that Shh per se affected H_2_O_2_ balance. Interestingly, untransfected neighbors of cells expressing Shh-mCherry also displayed increased H_2_O_2_ levels compared to neighbors of cells expressing only mCherry (Figure 6B). This observation suggests a paracrine effect of Shh on H_2_O_2_ levels. Moreover, when HyPer was addressed to the plasma membrane via the myristoylation and palmitoylation signal from the Lck tyrosine kinase (Lck-HyPer), filopodia bearing Shh were readily identified, and high levels of H_2_O_2_ consistently co-localized with Shh (Figure 6C). To quantitively assess the impact of Shh on H_2_O_2_ levels, we treated a stable cell line expressing Lck-HyPer with Shh-containing conditioned medium (Figure 6D). This treatment induced a global increase in H_2_O_2_ levels (Figure 6E,F), which was accompanied by filopodia growth, as quantified by their cumulative length (Figure 6G).

Finally, we investigated the molecular pathway used by Shh to increase H_2_O_2_ levels. Smo involvement was highly unlikely, as its mRNA was not detected in our RT-qPCR experiments (Appendix A). Indeed, cyclopamine (Shh-i), a Shh antagonist that specifically binds Smo, did not affect basal H_2_O_2_ levels and did not inhibit the Shh-induced H_2_O_2_ burst (Figure 6H,I). In contrast, treatment with Nox-i efficiently inhibited the Shh-mediated H_2_O_2_ increase, demonstrating the involvement of NADPH oxidase (NOX) enzymes (Figure 6I). Rac1 was an attractive candidate to bridge Shh and NOX in our system. It is ubiquitous, activates NOX2 (the only NOX complex present in HeLa cells, Appendix A), and is well known to regulate cytoskeleton dynamics [73]; in addition, its interplay with Shh and NOX1 was previously observed in a different system [74]. We used a Rac1 inhibitor (Rac1-i) to evaluate its involvement in H_2_O_2_ production induced by the Shh treatment. As shown in Figure 6J, Rac1-i itself had no effect on H_2_O_2_ levels in untreated cells but was able to block the effect of Shh treatment. Rac1 thus sits at the crossroads of Shh activity on HeLa cells, leading both to H_2_O_2_ production via the NOX/SOD (superoxide dismutase) complex and to the cytoskeleton remodeling that ultimately enables filopodium formation. A potential pathway to mediate Rac1 activation by Shh was recently published [75]: Shh-mediated axon guidance in the spinal cord depends on BOC receptor stimulation, leading to ELMO-Dock release and Rac1 activation. In addition, it was recently demonstrated that BOC is necessary for OPC formation in zebrafish [66] and we hypothesized it could be the receptor involved in the H_2_O_2_ enhancement by Shh. As BOC (Appendix A) and ELMO2 and 3 as well as several DOCK proteins (Human Protein Atlas [76]) are expressed in HeLa cells, we tested this hypothesis using a pan-Dock inhibitor (CPYPP, DOCK-i). As shown in Figure 6J, DOCK-i per se did not affect H_2_O_2_ levels but blocked the H_2_O_2_ increase induced by Shh. Moreover, simultaneous treatment with DOCK-i and Rac1-i had no cumulative effect (Figure 6J), suggesting that they belong to the same pathway. In summary, these experiments demonstrate that Shh induces a local increase in H_2_O_2_ levels in HeLa cells via a non-canonical route involving activation of Rac1, most likely via BOC and the Elmo-Dock pathway. As thoroughly discussed in Acevedo and Gonzalez-Billault [73], this leads to NOX2 activation along the redox branch of Rac1 activity and membrane protrusion growth along the actin cytoskeleton branch. Very recently, Shh cytoneme formation in rodent cells was demonstrated to depend on another feedback loop, including a Disp-BOC co-receptor complex [71].

## 4. Discussion

In the present study, we first proved that three main steps of Shh traffic are sensitive to physiological variations of H_2_O_2_ levels in Hela cells: primary secretion is inversely correlated to peroxide content while endocytosis and cell-to-cell transfer are directly correlated to it.

There are several possible ways by which H_2_O_2_ could affect these three steps. Our results demonstrated a clear accumulation of Shh in the Golgi apparatus upon increase in H_2_O_2_ levels, and this was not a general shutdown of the secretion machinery, as control proteins unrelated to Shh were still secreted to the same extent. Up to now and apart from the oxidative distress response in the ER, not much is known about how secretion might be influenced by physiological redox fluctuation. Two cases of non-conventional secretion have been demonstrated to be influenced by oxidation: periredoxins 1 and 2 (Prdx1, Prdx2) [77] and the transcription factor Engrailed-2 [41] and fusion of secretory vesicles during the conventional secretion of insulin was also shown to be affected [78], but the mechanisms are still elusive and the cellular contexts bear no clear relationship with the Shh primary secretion studied here. It remains to be determined whether Shh is preferentially blocked in the Golgi, or more actively retrieved from downstream compartments, and the mechanisms will deserve subsequent analysis.

Concerning the endocytic step, it has long been established that endocytosis may favor redox signaling in “redosoxomes” [79]; however, the converse effect of redox potential on endocytosis has been little studied, mostly in the context of distress [80,81] or in the context of plant physiology [82]. Of note, the thioredoxin-like protein TXNL1 [80] is able to convert redox changes into modification of GDI ability to capture Rab5, thus modulating endocytosis, and this represents an interesting hypothesis to study the mechanism of Shh endocytosis stimulation by H_2_O_2_.

Another possibility linking the effect of H_2_O_2_ signaling on Shh secretion and internalization would be a modification of either the expression of the receptor (Ptch) and/or co-receptors (Boc and Cdon) or their affinity for Shh. The change in expression cannot be precluded but is not the most likely explanation because the effect of H_2_O_2_ modulation on Shh traffic kinetics is much faster than the time needed for new receptor molecules to reach the cell surface. A change in the affinity of pre-existing receptors is an interesting hypothesis, given that Ptch, Boc, and Cdon (all expressed by HeLa cells, Appendix A) have been demonstrated to be endocytosed together with Shh, and found with Shh along filopodia [67,83].

The delivery step is itself a complex operation subdivided into distinct substeps, which could not be dissected in our experiments. Our results confirm that filopodia growth and density are sensitive to the redox state, but the exact place taken by filopodia and/or exosomes as well as the topology of delivery are still too uncertain to allow reasonable hypotheses on the mechanisms allowing its adjustment by H_2_O_2_. In particular, it will be necessary to clarify whether the transfer occurring at the end of filopodia takes place by the local budding of ectosomes, local release of exosomes, or by trogocytosis. In any case, cytoskeleton dynamics [84] and membrane proteins, such as integrins or disintegrins and metalloproteins [85], are already known to be susceptible to regulation by the oxidation level of their environment.

We extended these results by demonstrating that Shh distribution is also dynamic in vivo, at a time when the levels of H_2_O_2_ vary rapidly in time and space in the embryo (this study and [86]), and that experimentally modifying these levels modulates filopodia formation and leads to changes in the patterning of the central nervous system. Signaling filopodia or cytonemes are a common route for the secretion of morphogens [9,87], and it was recently demonstrated that several morphogens (Fgf2, Wnt8a, Wnt3a, Shh) are involved in a regulatory loop with cytoneme formation and growth, independent of their canonical pathway [71,88].

Finally, we discovered a feedback loop by which physiological levels of H_2_O_2_ adjust the trafficking of Shh, which, in turn, enhances H_2_O_2_ levels. It is worth noting that, in a different context (cross-talk between Shh and H_2_O_2_ during fin regeneration) we identified another type of feedback between Shh and H_2_O_2_ signaling, based on mechanisms completely unrelated to the BOC-Rac1 axis [17]. As a consequence of this feedback, local and timely H_2_O_2_ production may polarize the journey of Shh by modifying the rates of Shh secretion and endocytosis as well as the regulation of filopodia. We speculate that H_2_O_2_ could integrate the spreading of morphogens in a developing tissue, with different morphogens responding differentially to specific concentrations of H_2_O_2_ and, in turn, modifying H_2_O_2_ levels. Indeed, we already know that H_2_O_2_ impact on trafficking differs between Engrailed [41] and Shh (this study). This interplay between morphogens and redox signaling is also likely to have a role in various pathologies [19].

## 5. Conclusions

In a broader perspective, we propose that environmental insults or individual genetic variations inducing very subtle differences in H_2_O_2_ levels could impact morphogen distribution, resulting in inter-individual differences in organism patterning or disease susceptibility.

## Figures and Tables

**Figure 1 antioxidants-11-00718-f001:**
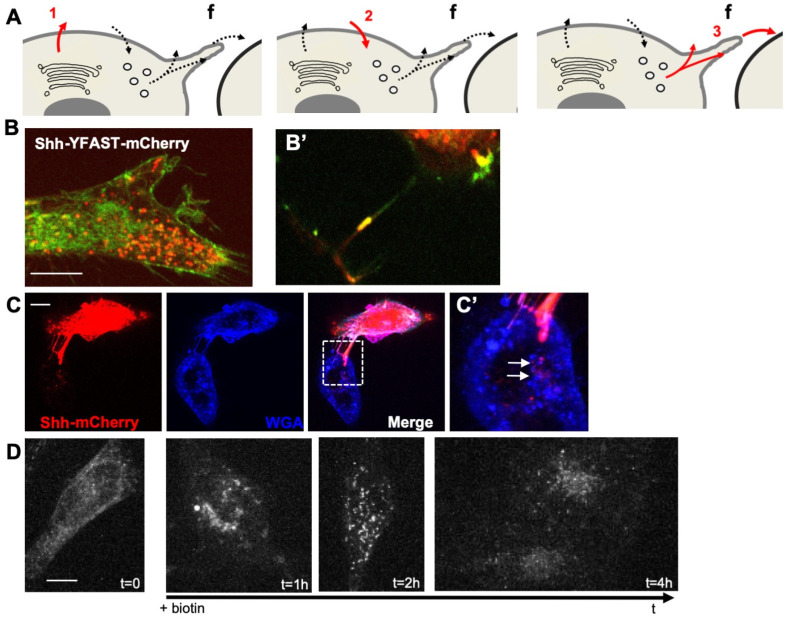
HeLa cells recapitulate typical events of Shh trafficking. (**A**) Schematic view of important steps analyzed in HeLa cells: (1) first secretion; (2) endocytosis; (3) dispatching to receiving cells; f: filopodium. (**B**) Double tagging with YFAST and mCherry allows the detection of Shh in three different compartments. Left panel: in a steady state, a cell producing Shh-YFAST-mCherry exhibits both a diffuse green signal (YFAST is detected early in the endoplasmic reticulum, when mCherry has not yet matured) and an abundant red vesicular signal (mCherry detected in the endosomes, where the pH prevents YFAST detection) and in filopodia where both tags are detected (**B’**). (**C**) In co-culture, pairs of Shh-producing (mCherry signal) and non-producing cells (both decorated by the binding of the WGA lectin to cell coat) often display filopodia between them. Large red dots in the non-producing cell indicate Shh transfer to recipient cells, and their position suggests delivery via the filopodia (white arrows in the enlarged inset in (**C’**)). (**D**) Shh-Sbp-mCherry secretion synchronized with the RUSH system displays classic timing: Shh is correctly hooked in the endoplasmic reticulum before biotin addition, reaches the Golgi 30–40 min after biotin addition, is secreted at approximately 1 h, can be easily detected within endosomes of the producing cell at 2 h, and can be visualized in non-producing cells at 4 h. Scale bars, 10 μm.

**Figure 2 antioxidants-11-00718-f002:**
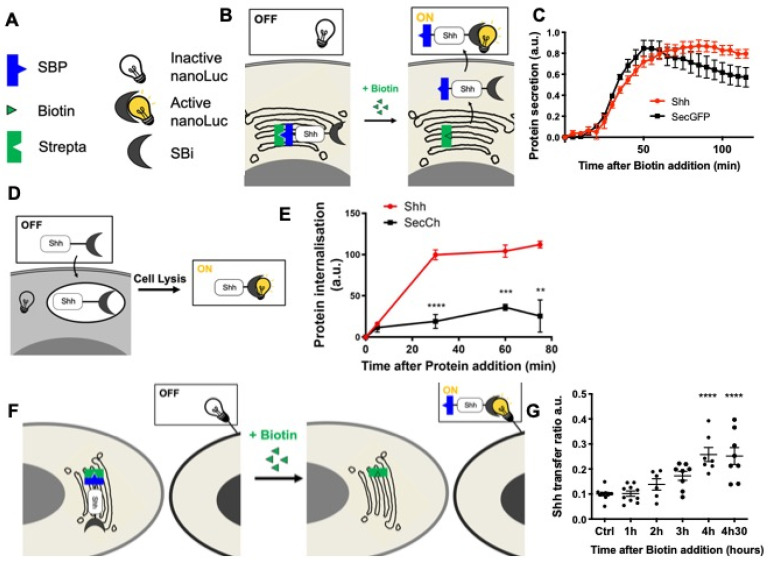
Quantitative monitoring of the journey of Shh in HeLa cells. (**A**–**C**) Quantifying Shh secretion by HeLa cells. (**A**,**B**) Schematic representation of the tools (**A**) and assay (**B**). (**A**) Strepta: core streptavidin linked to the KDEL hook; SBP: streptavidin-binding peptide linked to Shh or a secreted form of GFP (secGFP); SBi: small fragment of split nanoluciferase (HiBiT) linked to Shh or secGFP; inactive nanoLuc: large fragment of split nanoluciferase (LgBiT); active nanoLuc: reconstituted nanoluciferase. (**B**) Synchronized HeLa cells constitutively express the hook, and they express Shh fused to SBP and HiBiT (Shh-SBP-HiBiT) under doxycycline control. Purified LgBit added to the medium binds HiBiT when Shh reaches the extracellular space, and nanoLuc activity gives a measure of secretion. (**C**) Time course of Shh synchronized secretion compared to the control (secGFP fused to SBP and HiBiT). (**D**,**E**) Quantitating Shh internalization by HeLa cells. (**D**) Schematic representation of the assay. Shh fused to HiBiT (Shh-HiBiT), purified from the conditioned medium of producing HeLa cells and calibrated in vitro, was added to HeLa cells expressing inactive nanoLuc (LgBiT) in the cytosol. Upon cell lysis, nanoluciferase is reconstituted from endocytosed Shh-HiBiT and cytoplasmic LgBiT, allowing quantification of internalization. (**E**) Time course of Shh internalization compared to the control (a secreted form of mCherry: secCherry, fused to HiBiT and prepared in parallel to Shh). (**F**,**G**) Quantitating Shh delivery to recipient HeLa cells. (**F**) Schematic representation of the assay. Two HeLa cell populations are co-cultured. In the first one, synchronization of Shh release is achieved as in the secretion assay but for a longer period of time. The second population expresses inactive nanoLuc (LgBiT) at the cell surface (anchored via the CD4 transmembrane domain). Active nanoluciferase is reconstituted by the transport of tagged Shh (Shh-SBP-HiBiT) from a donor cell to the surface of a receiving cell. (**G**) Time-course analysis of Shh delivery to recipient cells becomes strongly significant approximately 3 h after biotin addition. Ctrl: without biotin. ** *p*-value ≤ 0.01; *** *p*-value ≤ 0.001; and **** *p*-value ≤ 0.0001. Details on statistics in Methods.

**Figure 3 antioxidants-11-00718-f003:**
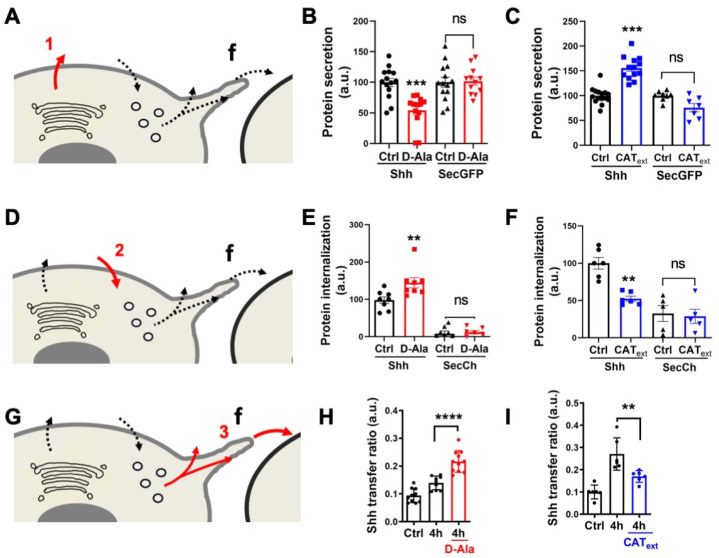
H_2_O_2_ affects Shh secretion, endocytosis, and delivery in HeLa cells. (**A**,**D**,**G**) Schematic steps: secretion (**A**), endocytosis (**D**), delivery (**G**); f: filopodium. A-C, Effects of increased (**B**) or decreased (**C**) H_2_O_2_ levels on secretion of Shh or secGFP from cells expressing Lck-DAO supplemented (or not) with D-Ala (**B**) or cells treated (or not) with extracellular catalase (CAT_ext_) (**C**). (**D**–**F**) Effects of increased (**E**) or decreased (**F**) H_2_O_2_ levels on Shh or secCh endocytosis by cells expressing Lck-DAO supplemented (or not) with D-Ala (**E**) or cells treated (or not) with CAT_ext_ (**F**). (**G**–**I**) Effects of increased (**H**) or decreased (**I**) H_2_O_2_ levels on delivery to recipient cell surface of cells expressing Lck-DAO supplemented (or not) with D-Ala (**H**) or cells treated (or not) with CAT_ext_ (**I**). ** *p*-value ≤ 0.01; *** *p*-value ≤ 0.001; and **** *p*-value ≤ 0.0001. Details on statistics in Methods.

**Figure 4 antioxidants-11-00718-f004:**
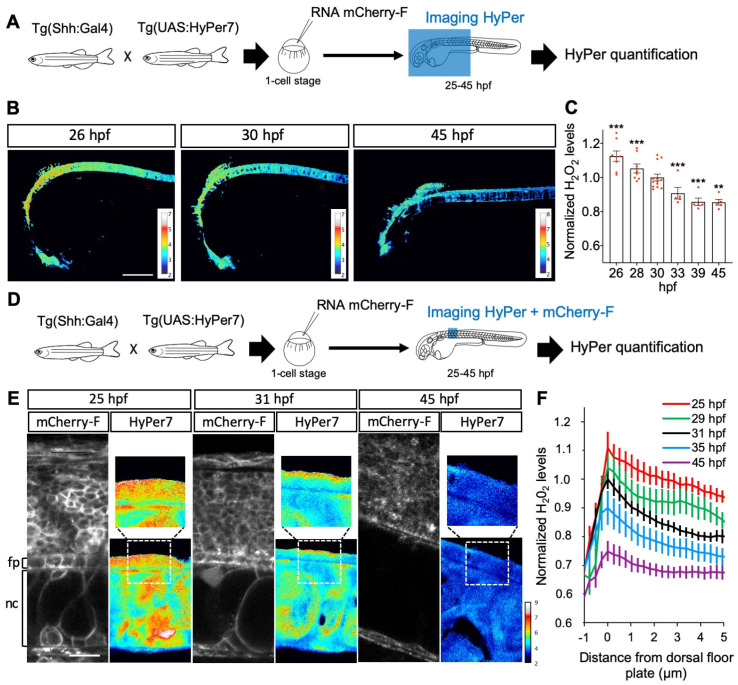
H_2_O_2_ levels are dynamic in time and space in the floor plate. (**A**,**D**) Experimental procedures for the results shown in B–C (**A**) and E–F (**B**). (**B**,**C**) Variation in time. (**B**) H_2_O_2_ levels in Tg(2.4Shha-ABC:Gal4-FF; UAS:Hyper7) zebrafish transgenic embryos at different times after fertilization as indicated (lateral view, anterior to the left). Scale bar, 200 μm. (**C**) Variations in H_2_O_2_ levels at different stages. Statistics are presented compared to 30 hpf. (**E**, **F**) Transient dorso-ventral gradient. (**E**) Embryos injected at the one-cell stage with mRNA for farnesylated mCherry (mCherry-F) were imaged for membranes (left panels: mCherry-F) and H_2_O_2_ (right panels: HyPer7; bottom: same field as in the left panel; top: enlarged view as indicated by the dotted line). fp: floor plate; nc: notochord. Scale bar: 20 μm. (**F**) Variations in H_2_O_2_ levels along the apico-basal axis of MFP cells. ** *p*-value ≤ 0.01 and *** *p*-value ≤ 0.001. Details on statistics in Methods.

**Figure 5 antioxidants-11-00718-f005:**
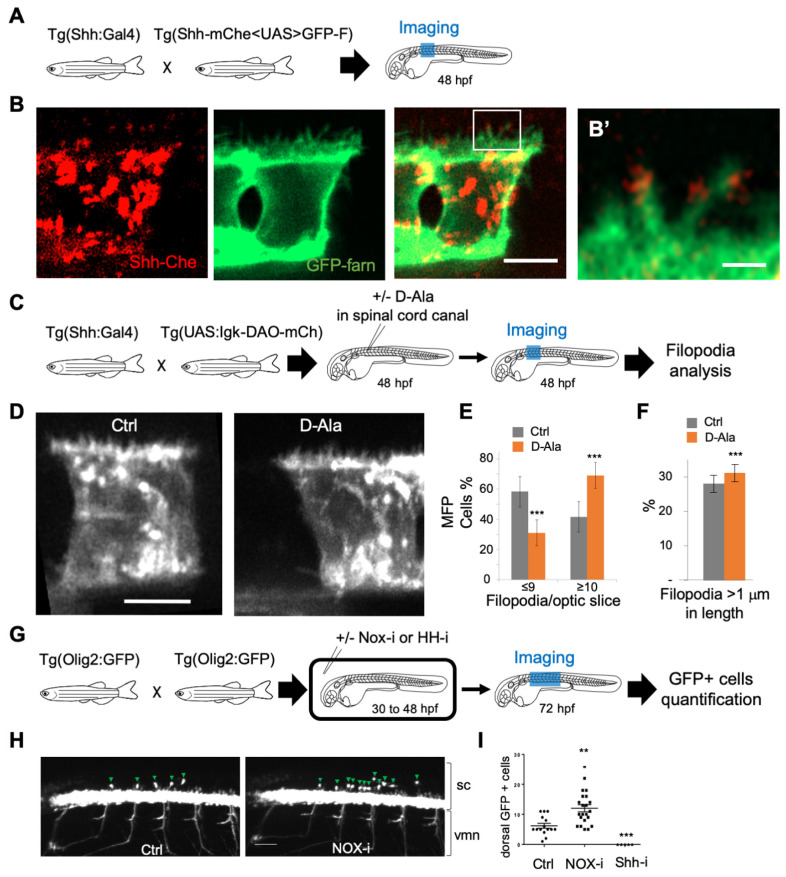
H_2_O_2_ modulates filopodia production and OPC behavior in the zebrafish spinal cord. (**A**,**C**,**G**) Experimental procedures for the results shown in B–B’ (**A**), D–F (**C**) and H–I (**G**). (**B**) Shh-mCherry visualization in MFP filopodia in live embryos at 48 hpf (Scale bar: 5 μm). (**B’**) Enlarged view (scale bar: 1 μm). (**D**) Visualization of filopodia at 48 hpf in Tg(2.4Shha-ABC:Gal4-FF;UAS:Igk-mb5-DAO-mCherry) before or after D-Ala (10 mM) injection in spinal cord canal (scale bar: 5 μm). (**E**) Quantification of filopodia in MFP cells (see Methods) in Ctrl and D-Ala-injected larvae. (**F**) Proportion of filopodia longer than 1 μm in MFPs of Ctrl and D-Ala-injected larvae. (**H**) Detection of GFP at 72 hpf in Tg(olig2:EGFP) larvae incubated from 30 to 48 hpf in control solution or in NOX-i at 10 μM. (**I**) Quantification of OPCs generated at 72 hpf after NADPH oxidase inhibitor (Nox-i) or cyclopamine (shh-i) treatments. ** *p*-value ≤ 0.01 and *** *p*-value ≤ 0.001. Details on statistics in Methods. sc: spinal cord; vmn: ventral motor nerves.

**Figure 6 antioxidants-11-00718-f006:**
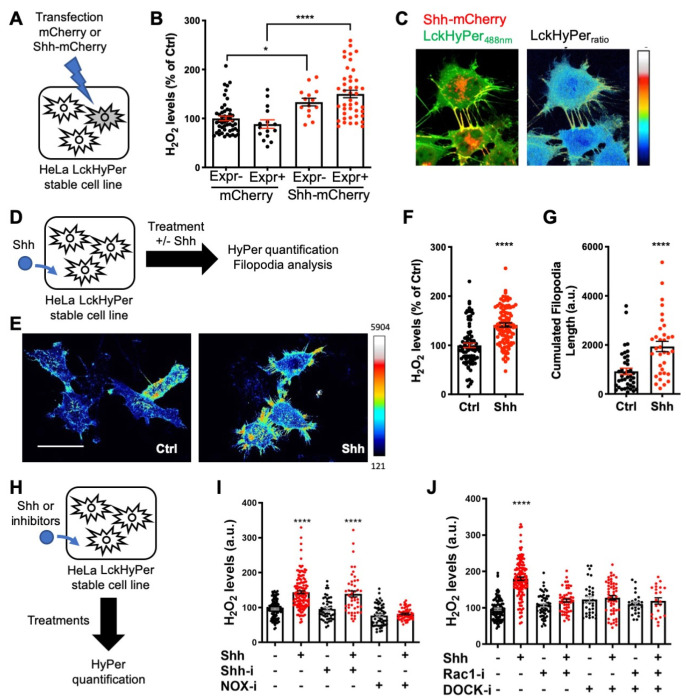
Shh regulates H_2_O_2_ levels and filopodial growth. (**A**,**D**,**H**) Experimental procedures for the results shown in B–C (**A**), E–G (**D**) and I–J (**H**). (**A**–**J**) Cells stably expressing membrane-bound HyPer (LckHyPer) either transfected with constructs coding for Shh-mCherry (**A**–**C**) or mCherry (**B**), were (**D**–**G**) treated (Shh) or not (Ctrl) with Shh, or were (**H**–**J**) treated with various inhibitors. (**B**) H_2_O_2_ was measured separately in cells expressing (Expr+) and non-expressing (Expr−) the transfected construct, coding for either Shh-mCherry (right) or mCherry (left). H_2_O_2_ levels in the non-expressing cells transfected with mCherry were taken as 100%. (**C**) Merging the red (Shh-mCherry) and green (Lck-Hyper_488nm_) signals highlights Shh-bearing filopodia (left), where H_2_O_2_ levels are distinctly elevated (right). (**E**–**G**) Treated cells (Shh) display higher H_2_O_2_ levels (imaged in (**E**), quantified in (**F**)) and have longer filopodia (imaged in (**E**), quantified in (**G**)) than untreated cells (Ctrl). (**I**) Cells treated (or not) with Shh were treated (or not) with either cyclopamine (Shh-i) or VAS2870 (NOX-i). (**J**) Cells treated (or not) with Shh were treated (or not) with either NSC23766 (Rac1-i) or CPYPP (DOCK-i) alone or in combination. In (**F**) and (**I**,**J**), untreated cells were taken as 100%. * *p*-value ≤ 0.05 and **** *p*-value ≤ 0.0001. Details on statistics in Methods. Scale bar: 50 μm.

## Data Availability

All of the data is contained within the article and the Appendix A.

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
