# Peer review of "Reciprocal Regulation of Shh Trafficking and H2O2 Levels via a Noncanonical BOC-Rac1 Pathway"

_antioxidants, 2022, doi:10.3390/antiox11040718_

Round 1

Reviewer 1 Report

  1. The abstract needs revision and highlights the outcome of the studies performed.
  2. In the introduction part, the authors need to emphasize on H2O2 and include the mechanism, how it regulates the HH proteins.
  3. The study lacks proper citations.
  4. The methodology section also lacks proper references.
  5. The abbreviations are not properly given.
  6. Again, in conclusion the authors should include the results of the study.

Author Response

  1. The abstract needs revision and highlights the outcome of the studies performed.

We have revised the abstract accordingly.

2. In the introduction part, the authors need to emphasize on H2O2 and include the mechanism, how it regulates the HH proteins.

As also requested by Reviewer 3, the introduction was supplemented with data and references on Shh traffic (one paragraph) and signaling pathways (one paragraph), and more information about the subcellular context of H2O2 production at the plasma membrane.

3. The study lacks proper citations.

We added references which indeed appear useful in the introduction and the discussion sections

4. The methodology section also lacks proper references.

Indeed, two references (only given in the "Results and discussion" section) were missing in the Methods section, where they have been added.

5. The abbreviations are not properly given.

The whole manuscript (including figure legends, figure themselves and supplementary material) has been carefully checked, and all acronyms are now identified and defined.

6. Again, in conclusion the authors should include the results of the study.

Indeed, we completely restructured the end of the manuscript, with a new separate section for discussion, incorporating explanations which were previously given in the "Results and discussion" section, more clearly stating the main results of our study, and discussing more thoroughly the various hypotheses which can be put forward to explain these results.

Reviewer 2 Report

In this manuscript, Thauvin at al. studied the regulation between the Shh production and trafficking in relationship with cellular levels of hydrogen peroxide and investigated the possible mechanism regulating this phenomenon in vitro. With the utilization of different techniques, some of them nicely set up by the authors, they followed the steps of production, secretion and internalization of Shh focusing on the transmission of this signal through filopodia. Using elegant zebrafish models and plasmid in HeLa cells they show how the modulation of H2O2 affect filopodia. In HeLa cells they show that inhibiting non-canonical Shh pathway through BOC RAC1, which increase the radicals production via NOX, the increase in H2O2 was abolished, concluding that the Shh stimulation of this pathway might have an important role in the Shh signaling.

Although most of the study was conducted with HeLa cancer cell line (which usage was clearly explained by the authors and has a valid rationale), and the in vivo data are partially following up their previous publication in Dev Cell, I think this is a quite elegant study with many point of interest that can be very useful for future studies.

The study is clearly described, the methods very detailed and the conclusion supports the data collected.

In my opinion and with my knowledge, some points need to be clarified by the authors and few experiments to be performed to clarify some of those:

  1. What´s WGA in Fig1.C? Are they nuclei?
  2. In Fig1.B The authors described that mCherry is localized in the endosomes. In D they write 1h in Golgi and 2h in endosome. The authors should double stain these compartments to claim these localizations. Moreover, can the author speculate why very little shh-mCherry is colocalized with YFAST close to the cellular membrane?
  3. Could you provide in the supplementary material more pictures as Fig.1 C´ where the mCHerry translocates in other neighboring cells?
  4. Related to Fig.3 It is unclear to me how catalase can enter in the cells and decrease the H2O2? The authors should measure the H2O2 as they did in figure 6 after catalase and D-ALA treatment to clarify the link with H2O2 of if the catalase is somehow acting extracellularly.
  5. Especially from data from figure 3 it seem that the feedback between secretion and internalization of shh is regulated in the way that less secretion might upregulate the receptor for shh. It would be important to show if these receptors are upregulated (or downregulated) when the shh secretion is decreased (or increased) via WB to understand if this is the case. Following up the pint number 4, is it possible that the catalase treatment alters ligand binding?
  6. HeLa cells express the primary cilium, which also during embryogenesis seems to be very important for development in zebrafish and other organisms. The primary cilium is a very important receiver of shh signaling and even if HeLa cells do not express smo, they express the shh receptor Ptch1 and 2. The authors do not touch on this subject because they described a non-canonical shh signaling. However the primary cilium might act as quencher of shh molecules and could also change morphology when they overexpress shh of inhibit the non-canonical pathway. Without asking to labell the cilia in zebrafish as the author did with the filopodia I would suggest to stain the primary cilium in Hela cells to describe: 1. If the shh-mCherry can be transferred directly there from one cell to the other (if technically possible). 2 to add the cilia length in correspondence of the treatment used in figures 6A (and place the cilia data in the supplementary).

I would also ask to the authors if they would think to draw the filopodia in a more appropriate way since by visualization it looks a primary cilium 

Author Response

  1. What´s WGA in Fig1.C? Are they nuclei?

WGA (Wheat Germ Agglutinin) is commonly used to label glycoproteins for imaging the cell surface in live or fixed cells, which we used to be sure of SHH localization. WGA and its function are now described in the text.

2. In Fig1.B The authors described that mCherry is localized in the endosomes. In D they write 1h in Golgi and 2h in endosome. The authors should double stain these compartments to claim these localizations. Moreover, can the author speculate why very little shh-mCherry is colocalized with YFAST close to the cellular membrane?

The difference between 1B and 1D stems from 1B analyzing Shh expression at steady state, whereas 1D analyzes a synchronized traffic. At steady state (1B) mCherry fluorescence is not detected in the secretion pathway because mCherry chromophore matures less rapidly than it takes for the molecule to reach the extracellular milieu. Conversely, YFAST does not fluoresce in endosomes because of acidic pH. Therefore, at steady state, green fluorescence corresponds to the secretion pathway (ER, Golgi and post Golgi), the red fluorescence corresponds to endosomes and colocalization of mCherry and YFAST fluorescence corresponds to the secondary secretion of Shh.

3. Could you provide in the supplementary material more pictures as Fig.1 C´ where the mCHerry translocates in other neighboring cells?

Unfortunately, the other pictures we have are of much lower quality

4. Related to Fig.3 It is unclear to me how catalase can enter in the cells and decrease the H2O2? The authors should measure the H2O2 as they did in figure 6 after catalase and D-ALA treatment to clarify the link with H2O2 of if the catalase is somehow acting extracellularly.

Catalase does not enter into the cells, but acts extracellularly. Indeed H2O2 is produced outside of the cell by the NOX/SOD3 complex before being reimported by aquaporins for signaling purposes. Consequently, extracellular catalase plays a role of sink for H2O2 as soon as it is produced.

This explanation was missing and has been added in the text.

5. Especially from data from figure 3 it seem that the feedback between secretion and internalization of shh is regulated in the way that less secretion might upregulate the receptor for shh. It would be important to show if these receptors are upregulated (or downregulated) when the shh secretion is decreased (or increased) via WB to understand if this is the case. Following up the pint number 4, is it possible that the catalase treatment alters ligand binding?

This is an interesting possibility which is now addressed in the new discussion section.

6. HeLa cells express the primary cilium, which also during embryogenesis seems to be very important for development in zebrafish and other organisms. The primary cilium is a very important receiver of shh signaling and even if HeLa cells do not express smo, they express the shh receptor Ptch1 and 2. The authors do not touch on this subject because they described a non-canonical shh signaling. However the primary cilium might act as quencher of shh molecules and could also change morphology when they overexpress shh of inhibit the non-canonical pathway. Without asking to labell the cilia in zebrafish as the author did with the filopodia I would suggest to stain the primary cilium in Hela cells to describe: 1. If the shh-mCherry can be transferred directly there from one cell to the other (if technically possible). 2 to add the cilia length in correspondence of the treatment used in figures 6A (and place the cilia data in the supplementary). I would also ask to the authors if they would think to draw the filopodia in a more appropriate way since by visualization it looks a primary cilium.

We agree with Reviewer 2 that the primary cilium acting as a sink, or the more unorthodox possibility that it could be a conduit for Shh transfer to recipient cells, are interesting hypotheses which would warrant experimental testing in the future.

Reviewer 3 Report

Summary:  In this manuscript, Thauvin et al. describe the effects of peroxide on sonic hedgehog (Shh) secretion, endocytosis, and transfer to nearby cells.  First, the authors describe several different approaches to track Shh release and reuptake in HeLa cells and, using these, report that peroxide inhibits secretion of Shh but enhances Shh endocytosis and transfer to adjacent cells.  In the second half of the paper, the authors move more in vivo to examine the role of peroxide on Shh secretory pathway events in zebrafish embryos, before returning to HeLa cells to monitor whether Shh participates in a feedback loop by altering peroxide levels.  In general, the manuscript is well-written, and the data are convincing.  With that said, the story is complicated by switching back and forth between different model systems and different types of assays with similar but distinct reporters; the model figures in the early part of the paper are helpful in understanding what is being monitored, but these are not included in the second half of the manuscript.  I recommend publication after the points below are addresses. 

Major concerns:

-The introduction seems very short and could better describe the Shh pathway or how peroxide levels can be increased during cell signaling events.  I recommend expanding this section by one or two paragraphs to ‘set the stage’ better for those reading.   

-The transition to the experiments in the zebrafish model needs improvement.  It is tough to follow how the experiments in Fig. 3 lead the authors to the experiments described in Figs. 4 and 5. 

-For Fig. 4 – When do peroxide levels peak post-fertilization?  It seems like levels increase significantly immediately post-fertilization followed by a decrease, but the timing of increase is not captured in the time course.  Either an additional experiment could be useful here or a reference to other papers documenting this increase, or (minimally) an explanation of why it has not been studied.  In addition, part of the figure has been cutoff at the right margin.  This needs to be corrected. 

-In reading, I missed a mechanistic proposal for how peroxide decreases Shh secretion from cells.  Some explanation for how this might happen seems warranted, with references to appropriate literature if available.

-A few schematics/simple models in the second half of the paper like those in the first half of the paper might be helpful for understanding the experiments.  I say this because many of the experiments the authors conduct are similar but with slight changes, and it is sometimes tough to follow what is being measured. 

Minor points/typos:

-For Figure 1 and the accompanying text, be sure to define what secGFP and secCherry are.  I assume these are secreted forms of GFP and mCherry fused to a domain of the split luciferase.

-Describe the utility of farnesylated GFP (GFP-farn) and mCherry-F in more detail in the text.  It is not apparent what these mean from the abbreviations and the text provided, although I assume they are just general plasma membrane markers. 

Author Response

Major concerns:

-The introduction seems very short and could better describe the Shh pathway or how peroxide levels can be increased during cell signaling events.  I recommend expanding this section by one or two paragraphs to ‘set the stage’ better for those reading.   

We have completely rewritten the introduction, as was also proposed by Reviewer 1.

-The transition to the experiments in the zebrafish model needs improvement.  It is tough to follow how the experiments in Fig. 3 lead the authors to the experiments described in Figs. 4 and 5. 

This transition has been better justified in the revised version.

-For Fig. 4 – When do peroxide levels peak post-fertilization?  It seems like levels increase significantly immediately post-fertilization followed by a decrease, but the timing of increase is not captured in the time course.  Either an additional experiment could be useful here or a reference to other papers documenting this increase, or (minimally) an explanation of why it has not been studied.  In addition, part of the figure has been cutoff at the right margin.  This needs to be corrected. 

During development, H2O2 peaks at somitogenesis and stays high until 48hpf with dynamic spatio-temporal reparation (Gauron et al 2016, Dev Biol 414(2) :133-141). The pictures were not cut off. With the objective used we could only image half the embryo.

-In reading, I missed a mechanistic proposal for how peroxide decreases Shh secretion from cells.  Some explanation for how this might happen seems warranted, with references to appropriate literature if available.

H2O2 induces a retention of SHH in the Golgi (Gauron et al 2016, Dev Biol 414(2) :133-141). We now present in the new "Discussion" section hypotheses about the possible mechanism of this inhibition.

-A few schematics/simple models in the second half of the paper like those in the first half of the paper might be helpful for understanding the experiments.  I say this because many of the experiments the authors conduct are similar but with slight changes, and it is sometimes tough to follow what is being measured. 

Schematics have been added to the figures

Minor points/typos:

-For Figure 1 and the accompanying text, be sure to define what secGFP and secCherry are.  I assume these are secreted forms of GFP and mCherry fused to a domain of the split luciferase.

This has been added (and more precisely defined both in the text, figure legends, material and methods, and supplementary table).

-Describe the utility of farnesylated GFP (GFP-farn) and mCherry-F in more detail in the text.  It is not apparent what these mean from the abbreviations and the text provided, although I assume they are just general plasma membrane markers. 

A clear definition of the farnesylation signal (indeed a farnesylation+palmitoylation signal from Ha-Ras addressing fused protein to the plasma membrane) and its function are now given both in the main text and in supplementary material.

Round 2

Reviewer 1 Report

The authors have now worked on all the comments and i expect its publication after the proof reading.

Author Response

Thank you very much for useful comments

Reviewer 2 Report

Thanks to the authors to have clarified some of the points I have raised. However, important points that could have been addressed by some suggested experiments have been left unanswered. Of special concern is the claim the transfer of Shh which has been only shown with a single picture.

These points have been left unanswered:

  1. What´s WGA in Fig1.C? The authors indicated that they add it in the text but it is not clear where to me.
  2. In Fig1.B The authors described that mCherry is localized in the endosomes. In D they write 1h in Golgi and 2h in endosome. The authors should double stain these compartments to claim these localizations.

Although the authors explain what should be the localization due to pH I think that the double staining is necessary to demonstrate this claim.

  1. Could you provide in the supplementary material more pictures as Fig.1 C´ where the mCHerry translocates in other neighboring cells?

More pictures need to be provided to support this claim.

  1. The authors describe that catalase acts extracellularly. However, the expression of DAO and H2O2 generation is intracellular. Thus is difficult to understand how catalase can improve the situation unless the H2O2 is secreted and reabsorbed. Thus, I think it is important to measure the H2O2 as they did in figure 6 after catalase and D-ALA treatment to clarify the link with H2O2 and catalase treatment as mentioned in the previous round.
  2.  Without asking to label the cilia in zebrafish as the author did with the filopodia I would suggest to stain the primary cilium in Hela cells to describe: 1. If the shh-mCherry can be transferred directly there from one cell to the other (if technically possible). 2 to add the cilia length in correspondence of the treatment used in figures 6A (and place the cilia data in the supplementary).

I would also ask to the authors if they would think to draw the filopodia in a more appropriate way since by visualization it looks a primary cilium

Author Response

These points have been left unanswered:

  1. What´s WGA in Fig1.C? The authors indicated that they add it in the text but it is not clear where to me.

However, we did explicitely mention (line 273 in the previous manuscript, line 274 in the present manuscript) that WGA is a lectin commonly used to decorate cell coat.

  1. In Fig1.B The authors described that mCherry is localized in the endosomes. In D they write 1h in Golgi and 2h in endosome. The authors should double stain these compartments to claim these localizations.

Although the authors explain what should be the localization due to pH I think that the double staining is necessary to demonstrate this claim.

Fig1.B is taken at steady state and Fig1D is a RUSH experiment, which means that distributions are expected to differ.

Golgi retention of Shh has already been proven (without mechanistic or kinetic analysis) by Bodipy TR ceramide staining (Gauron et al, 2016, Dev Biol, ref 16 in the revised manuscript). In addition, the new kinetic analysis permitted by the RUSH system allows compartment identification via the well calilbrated transit schedule of fluorescent proteins taken as controls (GFP, mCherry), as consistently established for the RUSH approach (Boncompain et al, 2012, Nat Meth, ref 60 in the revised manuscript).

Endocytosis of Hedgehog by source cells has been demonstrated in several different systems: fly, in many papers from Guerrero or Therond labs, also Parchure et al. (2015) MBoC 16:4700; insect cells Parchure (op cit); chicken Vyas et al., 2014, SciRep 4:7357; or mammalian cells Vyas (op cit).

On these grounds, and given the time constraints, we think reasonable to decline the suggestion.

  1. Could you provide in the supplementary material more pictures as Fig.1 C´ where the mCHerry translocates in other neighboring cells?

More pictures need to be provided to support this claim.

Additional pictures are provided for the Reviewer 2

  1. The authors describe that catalase acts extracellularly. However, the expression of DAO and H2O2 generation is intracellular. Thus is difficult to understand how catalase can improve the situation unless the H2O2 is secreted and reabsorbed. Thus, I think it is important to measure the H2O2 as they did in figure 6 after catalase and D-ALA treatment to clarify the link with H2O2 and catalase treatment as mentioned in the previous round.

We never performed an experiment using together extracellular catalase and DAO (an experiment which could not be meaningful). Extracellular catalase behaves as sink for H2O2 being transiently present in the extracellular milieu after production by NOX/SOD (not by DAO) and before its import across the plasma membrane via aquaporins. On the other hand, DAO mimics the intracellular rise of H2O2 subsequent to NOX activation, SOD3 constitutive activity, and import via aquaporins.

  1.  Without asking to label the cilia in zebrafish as the author did with the filopodia I would suggest to stain the primary cilium in Hela cells to describe: 1. If the shh-mCherry can be transferred directly there from one cell to the other (if technically possible). 2 to add the cilia length in correspondence of the treatment used in figures 6A (and place the cilia data in the supplementary).

I would also ask to the authors if they would think to draw the filopodia in a more appropriate way since by visualization it looks a primary cilium

To avoid misinterpretation, filopodium is now specified in the figures.
